# Monitoring Antibiotic Resistance in Wastewater: Findings from Three Treatment Plants in Sicily, Italy

**DOI:** 10.3390/ijerph22030351

**Published:** 2025-02-27

**Authors:** Roberta Magnano San Lio, Andrea Maugeri, Martina Barchitta, Giuliana Favara, Maria Clara La Rosa, Claudia La Mastra, Antonella Agodi

**Affiliations:** Department of Medical and Surgical Sciences and Advanced Technologies “GF Ingrassia”, University of Catania, 95123 Catania, Italy; robertamagnanosanlio@unict.it (R.M.S.L.); andrea.maugeri@unict.it (A.M.); martina.barchitta@unict.it (M.B.); giuliana.favara@unict.it (G.F.); mariaclara.larosa@unict.it (M.C.L.R.); claudia.lamastra@unict.it (C.L.M.)

**Keywords:** epidemiology, public health, surveillance, antimicrobial resistance, wastewater

## Abstract

Antimicrobial resistance (AMR) poses a global public health threat. Wastewater analysis provides valuable insights into antimicrobial resistance genes (ARGs), identifying sources and trends and evaluating AMR control measures. Between February 2022 and March 2023, pre-treatment urban wastewater samples were collected weekly from treatment plants in Pantano D’Arci, Siracusa, and Giarre (Sicily, Italy). Monthly composite DNA extracts were prepared by combining weekly subsamples from each site, yielding 42 composite samples—14 from each treatment plant. Real-time PCR analysis targeted specific ARGs, including *bla*SHV, *erm(A)*, *erm(B)*, *bla*OXA, *bla*NDM, *bla*VIM, *bla*TEM, and *bla*CTX-M. The preliminary findings revealed that *bla*ERM-B, *bla*OXA, *bla*TEM, and *bla*CTX-M were present in all samples, with *erm(B)* (median value: 8.51; range: 1.67–30.93), *bla*SHV (0.78; 0.00–6.36), and *bla*TEM (0.72; 0.34–4.30) showing the highest relative abundance. These results underscore the importance of integrating ARG data with broader research to understand the persistence and proliferation mechanisms of ARGs in wastewater environments. Future studies should employ metagenomic analyses to profile resistomes in urban, hospital, agricultural, and farm wastewater. Comparing these profiles will help identify contamination pathways and inform the development of targeted ARG surveillance programs. Monitoring shifts in ARG abundance could signal cross-sectoral contamination, enabling more effective AMR control strategies.

## 1. Introduction

The emergence of antimicrobial resistance (AMR) has become a global health crisis, with wastewater systems being identified as critical reservoirs for the proliferation and dissemination of antibiotic resistance genes (ARGs). Wastewater treatment plants (WWTPs) often receive domestic, industrial, and hospital waste, which may carry high concentrations of antibiotics, microorganisms, and ARGs [1,2,3]. When discharged untreated or partially treated, wastewater can release ARGs into the environment, where they may reach natural water bodies, impacting ecosystems and potentially entering the human water supply. The need for accurate monitoring of ARG prevalence and trends in urban wastewater is essential for public health risk assessments and environmental protection [4]. In this context, wastewater-based epidemiology (WBE) is an epidemiological approach that utilizes the analysis of wastewater to gather insights about human populations at a community-wide level [5]. The effective monitoring of ARGs involves sampling from rivers, lakes, and wastewater outlets and using advanced molecular techniques, such as quantitative PCR (qPCR) and metagenomic sequencing, to identify and quantify ARGs in situ. This approach provides comprehensive data on the ARG types, concentrations, and potential hotspots, revealing trends and possible links to human activities, antibiotic usage patterns, and seasonal variations [6,7]. Consistent ARG surveillance offers valuable insights into the ecological and public health impacts of AMR in water systems. Monitoring supports targeted actions, such as refining WWTP processes to limit ARG discharge and setting up early-warning systems in high-risk areas. These insights not only inform environmental protection strategies but also bolster public health efforts by enabling timely interventions, ultimately helping to limit the spread of waterborne, antibiotic-resistant pathogens [8]. Prominent ARGs, such as *bla*SHV, *bla*OXA, *bla*NDM, *bla*VIM, *bla*TEM, *bla*CTX-M, and *erm(A)* and *erm(B)*, confer resistance to crucial antibiotic classes, including beta-lactams and macrolides [9]. The *bla* genes, for instance, provide resistance to beta-lactams by encoding beta-lactamase enzymes that break down these antibiotics, with *bla*CTX-M, *bla*NDM, *bla*VIM, and *bla*OXA being associated with extended-spectrum and carbapenem-resistant infections [10]. Similarly, the *erm(A)* and *erm(B)* genes lead to macrolide resistance in bacteria, complicating the treatment of infections caused by pathogens like *Staphylococcus* and *Streptococcus* species [11]. These ARGs are often carried on mobile genetic elements such as plasmids, which facilitates their rapid spread across bacterial populations and environments, from wastewater and soil to clinical settings [12,13]. The presence of these ARGs in both healthcare and environmental reservoirs exacerbates public health risks by making infections harder to treat, increasing patient mortality, prolonging hospital stays, and driving up treatment costs. This highlights the need for robust surveillance and containment strategies to curb the spread of these highly transmissible and impactful ARGs [14]. The present study investigates the presence and abundance of ARGs within three WWTPs located in Pantano D’Arci (Catania, Sicily, Italy), Siracusa (Sicily, Italy), and Giarre (Catania, Sicily, Italy), which represent varied geographical and demographic profiles. By using real-time PCR as the molecular technique, the study precisely quantifies clinically relevant ARGs. The aim is to quantify the relative abundance of the most common ARGs, contributing to a broader understanding of antibiotic resistance spread through urban wastewater. These results will then be compared with those obtained from an alternative, cost-effective method, namely a DNA-based electrochemical biosensor.

## 2. Materials and Methods

### 2.1. Sample Collection and Handling

Wastewater samples were collected weekly from three WWTPs between February 2022 and March 2023. Wastewater samples were collected at the beginning of each week for practical and logistical reasons. Sampling occurred at the inlet, before any treatment process. The sampling sites included Pantano D’Arci (Catania, Sicily, Italy), Siracusa (Sicily, Italy), and Giarre (Catania, Sicily, Italy), which vary in population size and wastewater characteristics. Each sample (100 mL) was stored in a polyethylene bottle with secure caps and labeled to indicate the sampling site and date. The samples were transported under refrigerated conditions, maintaining a temperature range of 5 ± 3 °C, to prevent nucleic acid degradation during transit. Upon arrival at the laboratory, each sample was divided into two aliquots of 50 mL. One aliquot was immediately frozen for future confirmatory testing, while the other was processed for ARG analysis.

### 2.2. Sample Processing and Concentration

The process of concentrating wastewater samples followed a series of preparation steps designed to isolate microbial components, as outlined in the protocol described in [15]. Starting with 45 mL of each sample, a pre-treatment step in a 56 °C water bath was conducted to reduce microbial activity. This was followed by rapid cooling and centrifugation at +4 °C to separate particulates. Additional centrifugation steps, combined with the addition of polyethylene glycol (PEG 8000) and sodium chloride (NaCl), facilitated the precipitation of microbial cells. After two hours of centrifugation, a visible pellet formed, which was used for the subsequent nucleic acid extraction. This pellet contained concentrated microbial DNA, essential for accurate ARG detection in subsequent analyses.

### 2.3. Nucleic Acid Extraction

To ensure consistent extraction of nucleic acids from each sample, the eGENE-UP platform (bioMerieux, Marcy-l’Étoile, France) was employed, as referenced in [15]. This system uses magnetic silica particles to bind nucleic acids, with the lysis being facilitated by guanidine thiocyanate. This lysis step effectively disrupts microbial cells, denaturing proteins, RNases, and DNases to prevent nucleic acid degradation. Following incubation, silica particles were added to the mixture, capturing nucleic acids. A series of washing steps then removed residual sample components, and a final elution with a TE buffer (pH 8.0) ensured that the nucleic acids were in optimal condition for molecular analysis. This method yields high-quality DNA, which is necessary for accurate real-time and digital PCR analysis.

### 2.4. DNA Pooling for Composite Samples

To account for daily and weekly fluctuations in wastewater composition, we created monthly composite DNA samples. Composite sampling is a widely recommended approach in environmental monitoring [16]. In general, this method provides a more representative snapshot of microbial and genetic profiles over time. Specifically, we collected one sample per week and combined the four weekly samples into a single monthly extract. At each of the three WWTPs, the weekly samples from four consecutive weeks were pooled to capture the monthly ARG profile. Over the 14-month study period, this process generated 14 pooled samples per site, totaling 42 across all locations. By integrating multiple time points, this approach smooths out short-term fluctuations, offering a more stable measure of ARG dynamics and improving the assessment of long-term trends in antibiotic resistance.

### 2.5. Real-Time PCR

The QuantStudio™ 7 Flex Real-Time PCR System (Applied Biosystems, Foster City, CA, USA) was used to measure the relative abundance of ARGs in each sample, following the manufacturer’s protocol [17]. All the analyses were conducted in triplicate for each composite DNA sample. Target genes included *bla*SHV, *bla*OXA, *bla*NDM, *bla*VIM, *bla*TEM, *bla*CTX-M, *erm(A)*, and *erm(B).* Each gene’s abundance was normalized against the bacterial 16S rRNA gene, serving as an internal reference. Specifically, the total volume per reaction was 18 µL, with 10.0 µL of TaqPath™ qPCR Master Mix, CG, 5 µL of TaqMan^®^ Gene Expression Assay, and 3 µL of nuclease-free water per reaction. All reactions were performed in triplicate. The standard thermal cycling profile included 2 min of UNG incubation at 50 °C, 20 s of Polymerase activation at 95 °C, and 40 cycles of PCR with 15 s of denaturation at 95 °C and 1 min of anneal/extend at 60 °C. Using the ΔCT method, relative abundances were calculated with the formula 2^(−ΔCT)^, where ΔCT represents the difference between the target gene’s CT value and the CT of the 16S rRNA reference.

### 2.6. Statistical Analysis

The statistical analyses were performed using SPSS (version 26). Descriptive statistics, such as medians, ranges, and percentages, were employed to summarize the prevalence and relative abundance of ARGs, providing a comprehensive overview of the data. Heatmaps were created to visually illustrate the distribution of ARG concentrations across the three WWTP sites and throughout the sampling periods, facilitating the identification of spatial and temporal patterns. Differences in ARG abundance among the three WWTPs were analyzed and represented through box plots, highlighting variability and site-specific trends. Temporal trends in ARG abundance were effectively evaluated and depicted using line graphs, offering a clear visualization of changes over time.

## 3. Results

The analysis included untreated weekly wastewater samples, collected between February 2022 and March 2023, which were subsequently combined into 42 monthly composite samples. These composite samples were derived from a total of 168 weekly samples (42 months × 4 weeks per month), where portions from the weekly samples were combined to create a representative monthly composite for each site. This approach ensured that the composite samples accurately represented the conditions across the entire month, allowing for a comprehensive assessment of the ARG prevalence and abundance across all locations. Samples were collected from three wastewater treatment plants (WWTPs) in the provinces of Catania (Pantano D’Arci and Giarre) and Siracusa (Figure 1). The Pantano D’Arci plant serves the municipality of Catania, accommodating approximately 300,000 residents. The Giarre plant provides treatment for the municipalities of Giarre, Mascali, and Riposto, covering a population of around 57,000. Meanwhile, the Siracusa plant serves the municipalities of Siracusa, Floridia, and Solarino, with nearly 150,000 residents. Both the Pantano D’Arci and Siracusa facilities utilize tertiary treatment, an advanced process that includes additional filtration and disinfection steps. In contrast, the Giarre plant employs secondary treatment, which primarily relies on biological processes to remove organic matter.

In general, the results demonstrated a high frequency of ARGs across the examined 42 monthly composite wastewater samples. Real-time PCR data were normalized against the 16S rRNA control gene and calculated using the 2^(−ΔCT)^ method, with results being expressed after scaling by a factor of 10^5^ for clarity. The relative abundance of each ARG was visualized through a heatmap (Figure 2), which represented the gene concentrations across the various samples and sites, allowing for an effective comparison of their prevalence. More specifically, *erm(B)*, *bla*OXA, *bla*TEM, and *bla*CTX-M were detected in 100% of samples, although with differing relative abundances, highlighting their constant presence across all sites and sampling times. *bla*CTX-M was particularly abundant and was confirmed with multiple probes, with detection rates for individual probes varying between 45.2% and 100%. The *bla*CTX-M gene is commonly associated with resistance to beta-lactam antibiotics, making its ubiquitous detection especially concerning. Other notable ARGs included *bla*SHV, which was detected in 95.2% of samples and is often linked to nosocomial infections, suggesting the presence of potentially pathogenic strains in wastewater. The *erm(A)* gene was present in 81% of samples, indicating ongoing exposure to macrolide antibiotics, which are commonly used in both human and veterinary medicine. *bla*NDM was identified in 69% of the samples with one probe and 57.1% with another, reflecting the occurrence of carbapenem-resistant bacteria—a critical concern given the role of carbapenems as last-line antibiotics. Furthermore, *bla*VIM was detected in 97.7% of the samples, emphasizing its widespread prevalence across different treatment sites.

Figure 3 provides a detailed overview of the relative abundances of the ARGs, considering the three reference sites collectively. The ARGs with the highest relative abundances were *erm(B)*, *bla*SHV, and *bla*TEM, which aligns with earlier findings (Figure 3A). For instance, *erm(B)* showed a median value of 8.51 (range: 1.67–30.93), followed by *bla*SHV with a median of 0.78 (range: 0.00–6.36) and *bla*TEM with a median of 0.72 (range: 0.34–4.30). Additional ARGs such as *erm(A)*, *bla*OXA, *bla*NDM, and *bla*CTX-M showed lower median values, indicating varying levels of prevalence and potential hotspots of antibiotic resistance (Figure 3B).

When analyzing ARGs by individual sites, differences in relative abundances became evident. Likewise, the results were presented separately for the most and least abundant ARGs (Figure 4A and 4B, respectively). The Pantano D’Arci (Catania) site exhibited elevated levels of *bla*OXA, along with *erm(B)*, *bla*SHV, and *bla*TEM, which were consistently abundant across all three sites. Siracusa and Giarre also showed similar trends but with some variation in the abundance of less prevalent ARGs. Notably, *bla*OXA was more prevalent at the Catania site.

Finally, we focused on the most abundant ARGs (i.e., *erm(B)*, *bla*SHV, *bla*TEM, and *bla*OXA) across the three sites, examining their temporal trends (Figure 5). The results generally revealed that the ARG abundance peaked in September and October 2022 across all sites. Specifically, *erm(B)* showed elevated relative abundances in Catania and Giarre during these months, while Siracusa showed an increase from September to March 2023 (Figure 5A). For *bla*SHV, a consistent increase was observed at all sites in September and October 2022, with Siracusa maintaining a stable pattern until a slight rise in early 2023 (Figure 5B). The *bla*TEM gene exhibited a peak in Catania in September, followed by a gradual decline, whereas in Siracusa, the relative abundances remained stable until an increase in March 2023. At Giarre, *bla*TEM reached its peak in November 2022 (Figure 5C). *blaOXA* exhibited increased prevalence in Catania from August to October 2022, while Giarre showed a similar but less pronounced peak in July and October 2022. In contrast, its levels remained relatively stable in Siracusa (Figure 5D).

## 4. Discussion

These results provide valuable insights into the prevalence and patterns of ARGs in urban wastewater before treatment, underscoring the potential public health risks that are posed by the spread of antibiotic resistance in community settings. Specifically, the *bla*SHV—Sulphydryl Variable Beta-Lactamase—gene encodes a class A beta-lactamase enzyme that hydrolyzes a wide range of beta-lactam antibiotics, including penicillins and some cephalosporins. SHV-type enzymes are commonly associated with antibiotic resistance in clinical isolates, particularly in *Enterobacteriaceae.* This gene is a key indicator of resistance in healthcare settings, where infections with SHV-producing bacteria are often difficult to treat [18]. The *bla*OXA—Oxacillinase Beta-Lactamase—gene encodes a class D beta-lactamase that is known for its resistance to oxacillin and other beta-lactam antibiotics. OXA-type enzymes are often found in *Acinetobacter* species and some *Enterobacteriaceae*, contributing to resistance to carbapenems, a critical antibiotic class. *bla*OXA has become a major target in surveillance programs, given its prevalence in hospital-acquired infections [18]. The *bla*NDM—New Delhi Metallo-Beta-Lactamase—gene is a class B metallo-beta-lactamase that provides resistance to a broad range of beta-lactam antibiotics, including carbapenems. Originally identified in New Delhi, this gene has since spread globally and is associated with multidrug-resistant infections, particularly in *Escherichia coli* and *Klebsiella pneumoniae*. *bla*NDM is particularly concerning due to its ability to confer resistance to last-resort antibiotics [19]. The *bla*VIM—Verona Integron-Mediated Metallo-Beta-Lactamase—gene encodes another class B metallo-beta-lactamase that provides resistance to carbapenems and other beta-lactam antibiotics. VIM-type enzymes are often found in *Pseudomonas* and *Enterobacteriaceae* species. The widespread presence of *bla*VIM in clinical and environmental samples raises concerns about its role in limiting treatment options for resistant bacterial infections [20]. The *bla*TEM—Temoneira Beta-Lactamase—gene encodes a class A beta-lactamase enzyme that is commonly associated with resistance to penicillins and early-generation cephalosporins. TEM-type enzymes are widespread among Gram-negative bacteria and are often found on mobile genetic elements, enabling their transfer between bacterial species. *bla*TEM is highly prevalent in both clinical and environmental settings [21]. The *bla*CTX-M—Cefotaximase Beta-Lactamase—gene encodes a class A extended-spectrum beta-lactamase (ESBL), which provides resistance to cephalosporins, including cefotaxime. CTX-M enzymes are among the most common ESBLs worldwide and are significant in hospital and community settings. The gene is prevalent in *Escherichia coli* and *Klebsiella pneumoniae*, frequently causing resistant infections in humans [22]. The *erm(A)* and *erm(B)*—Erythromycin Ribosome Methylase—genes encode enzymes that methylate the bacterial ribosome, conferring resistance to macrolide antibiotics, such as erythromycin, by preventing the antibiotic from binding effectively. These genes are prevalent in both human and animal microbiomes and are a significant concern in veterinary and agricultural settings, as they contribute to cross-resistance between human and animal pathogens [23].

In our study, the consistent detection of high-abundance genes, particularly *erm(B)* and *bla*TEM, across all sites and sampling times reflects the stable integration of these ARGs into the microbial population within wastewater. This persistence aligns with the current literature, which links elevated ARG concentrations in wastewater to substantial antibiotic use in the community and the presence of resistant bacterial strains [24,25]. Given their role in resistance to macrolides and beta-lactams—two widely used antibiotic classes—these genes could represent a baseline of ARGs that circulate in the community, facilitated by the frequent discharge of resistant bacteria from human and animal sources into wastewater systems [26].

The detection of *bla*OXA and *bla*CTX-M, genes that are frequently associated with hospital-acquired infections, emphasizes the role of wastewater as a reservoir for clinically relevant ARGs. These ARGs, often found in hospital environments where antibiotic usage is particularly high, suggest the potential flow of resistant pathogens from healthcare settings into public wastewater systems [27]. Their consistent presence, especially at the Pantano D’Arci site, raises questions about how specific environmental or operational conditions at treatment plants—such as the temperature, pH, and treatment efficacy—may influence the survival and persistence of certain ARGs. This finding opens up hypotheses around the influence of wastewater treatment conditions on the evolution and maintenance of ARG diversity. For instance, higher temperatures or suboptimal pH levels might favor the growth of specific microbial communities that harbor these resistance genes, supporting their persistence and even promoting horizontal gene transfer within bacterial populations. The statement about how higher temperatures or suboptimal pH levels can promote the growth of microbial communities harboring antibiotic resistance genes and facilitate horizontal gene transfer is supported by research highlighting environmental factors that influence microbial ecology and genetic exchange. Studies have shown that stressors like temperature changes or shifts in pH can create selective pressures that enhance the stability of resistant strains within a microbiome, particularly when resistance genes are mobile, such as those on plasmids or transposons. These conditions may also encourage horizontal gene transfer mechanisms, enabling the spread of resistance genes across bacterial populations [28,29].

The seasonal peak in ARG abundance observed in late 2022 introduces additional potential relations. This peak may reflect increased antibiotic consumption during certain times of the year, such as the winter months when respiratory infections and other bacterial infections typically lead to higher numbers of antibiotic prescriptions [30]. Alternatively, fluctuations in ARG levels may also correlate with population density changes, such as those that are seen in areas with seasonal tourism, which could temporarily increase the bacterial load in wastewater [31]. During the study period, *erm(B)* showed higher relative abundances in Catania and Giarre, while Siracusa experienced an increase from September to March 2023. Similarly, *blaSHV* levels rose at all sites in September and October 2022, with Siracusa maintaining a steady trend until a slight increase in early 2023. For *blaTEM*, a peak was observed in Catania in September, followed by a gradual decline. In Siracusa, the levels remained stable before increasing in March 2023, while in Giarre, the highest abundance was recorded in November 2022. As for *blaOXA*, its prevalence increased in Catania between August and October 2022, whereas Giarre showed a similar but smaller peak in July and October. In contrast, Siracusa exhibited a relatively stable pattern throughout the period. These findings suggest that climatic conditions, wastewater composition, and possibly fluctuations in antibiotic use could be driving the temporal trends observed. Further investigation into such seasonal trends could provide actionable insights, suggesting that public health interventions targeting antibiotic use during specific periods might help mitigate the amplification of ARGs in wastewater.

These findings underscore the utility of WBE as an invaluable tool for tracking public health risks related to antimicrobial resistance. Regular ARG monitoring could serve as an early warning system for emerging outbreaks of antibiotic-resistant infections, allowing for timely public health responses and the potential to implement targeted interventions in affected communities [32]. Additionally, this approach could inform treatment plant optimization by identifying conditions that minimize the persistence and spread of ARGs. Future research could integrate additional metadata, such as regional antibiotic prescription rates, specific resistance mechanisms, and the real-time operational conditions of the treatment plants (e.g., hydraulic retention time, organic load), to enhance the interpretative power of these findings. By providing a more detailed view of how ARGs circulate in wastewater, such research could inform both local and global strategies aimed at reducing the spread of antibiotic resistance, ultimately contributing to the safeguarding of public health [33].

The study provides significant insights into the prevalence and patterns of ARGs in urban wastewater, presenting a powerful basis for understanding public health risks related to antibiotic resistance. A key strength is the consistent detection of high-abundance ARGs, such as *erm(B)* and *bla*TEM, across various sites and times, suggesting a stable microbial community carrying these genes in wastewater. This consistency aligns with the existing literature on community antibiotic use, supporting the idea that wastewater can serve as a reliable proxy for community antibiotic resistance levels. Additionally, the study’s focus on WBE highlights its value as an early-warning tool for potential outbreaks of antibiotic-resistant infections, enabling timely public health interventions. The inclusion of ARGs associated with hospital environments, such as *bla*OXA and *bla*CTX-M, emphasizes wastewater’s role as a reservoir of clinically significant resistance genes, making this study particularly relevant for assessing public health threats. Moreover, the observed seasonal trends in ARG abundance may provide actionable insights into the impact of fluctuating antibiotic usage and population densities on ARG patterns, potentially guiding targeted public health measures. The potential to optimize wastewater treatment conditions based on ARG monitoring data further underscores the applied value of the study.

Despite these strengths, several limitations should be considered when interpreting the results. First, the study’s reliance on wastewater samples limits its ability to identify specific sources of ARGs, such as hospitals, households, or industrial contributors, reducing the specificity of potential public health interventions. Moreover, a potential limitation of using monthly composite samples is the loss of short-term variability, as this approach averages fluctuations that may occur on a daily or weekly basis. As a result, transient peaks or sudden changes in ARG abundance might not be fully captured, potentially affecting the detection of rapid shifts in microbial composition or contamination events. Another limitation is the lack of data on antibiotic prescription rates, treatment plant operating conditions, and population demographics, which could provide further context for the detected ARG patterns and seasonal fluctuations. Additionally, while high-abundance genes were consistently detected, low-abundance or emerging ARGs may be underrepresented due to detection limits, potentially obscuring newly developing resistance trends. The study also does not address potential differences in ARG persistence and survival across various wastewater treatment processes, which could provide insights into treatment optimization strategies. Moreover, we did not collect post-treatment samples in the early phase of the study, and this has resulted in the inability to compare pre- and post-treatment samples to assess the effectiveness of the treatment plant. However, further studies are ongoing and will provide this information in the future. Finally, the observed seasonal trends, while suggestive, lack direct correlations with specific environmental or social factors, limiting the ability to draw concrete conclusions about the drivers of ARG fluctuations over time. Future research with comprehensive metadata integration and expanded sampling could address these gaps, enhancing the accuracy and applicability of ARG surveillance in wastewater systems.

## 5. Conclusions

Our findings highlight the need to contextualize ARG data within the broader scope of existing research by obtaining valuable information on the main characteristics of prevalent ARGs and the operational conditions of wastewater treatment plants. Such contextualization will allow for a deeper understanding of our results and enable the development of hypotheses regarding potential mechanisms behind ARG persistence and proliferation in wastewater environments. Future work should include metagenomic analyses to characterize the resistome profile in the wastewater samples collected. Additionally, establishing comprehensive metagenomic profiles for urban and/or hospital wastewater—which may overlap—and comparing them to profiles from agricultural runoff and farm wastewater will be essential. This approach may allow for the development of tailored ARG surveillance programs, particularly by identifying shifts in ARG abundance that could signal cross-sectoral contamination.

## Figures and Tables

**Figure 1 ijerph-22-00351-f001:**
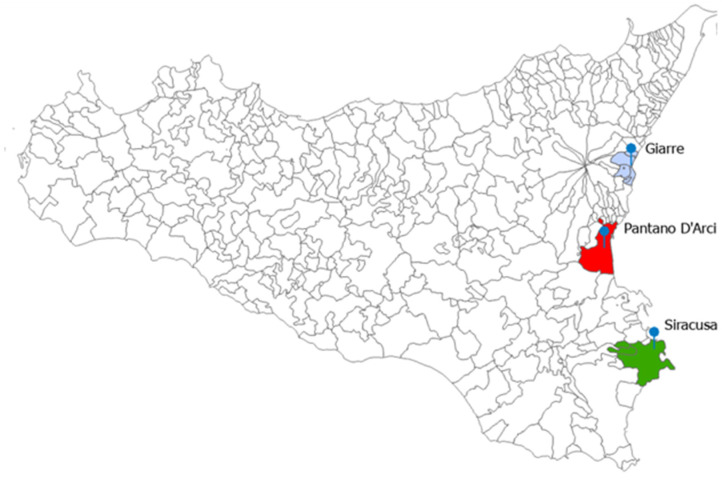
Map of the Sicily region displaying municipal boundaries (NUTS-3 level) and the locations of the wastewater treatment plants that were analyzed in this study, along with their respective service areas. Light blue represents municipalities served by the Giarre plant, red indicates the municipality of Catania served by the Pantano D’Arci plant, and green denotes municipalities served by the Siracusa plant. The map was created using ArcGIS Pro (v. 3.4).

**Figure 2 ijerph-22-00351-f002:**
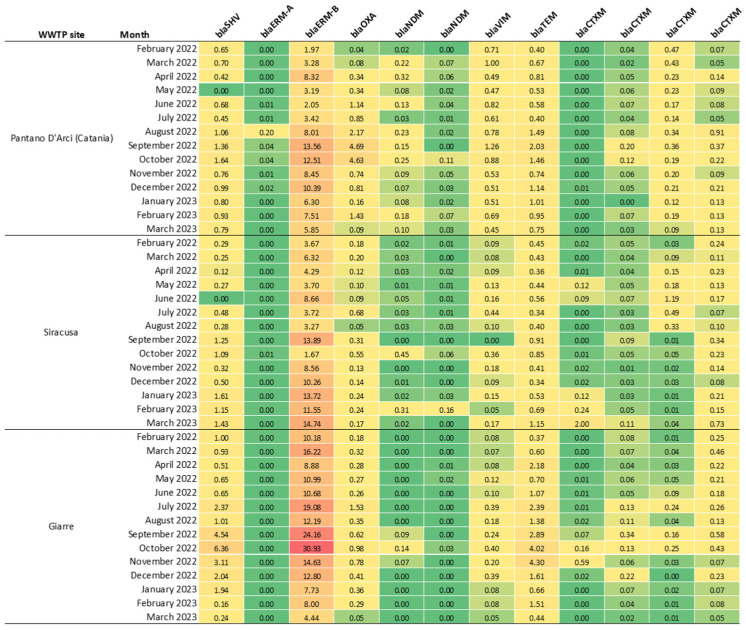
Relative abundances of ARGs in composite wastewater samples. Multiple probes were used for the *blaNDM* and *blaCTXM* genes. The figure illustrates relative abundance from low (red) to high (green) values.

**Figure 3 ijerph-22-00351-f003:**
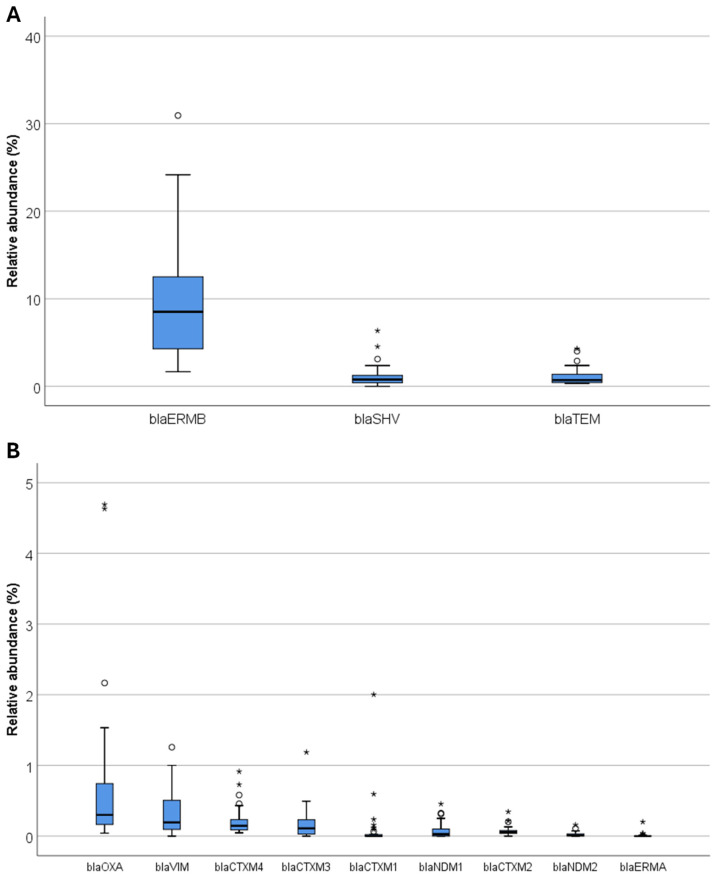
Relative abundances of ARGs. (**A**) highlights the most abundant ARGs, including *erm(B)*, *bla*SHV, and *bla*TEM, while (**B**) displays ARGs with lower relative abundances. The figures present box plots of relative abundances for each ARG, with circles indicating potential outliers and asterisks representing extreme values.

**Figure 4 ijerph-22-00351-f004:**
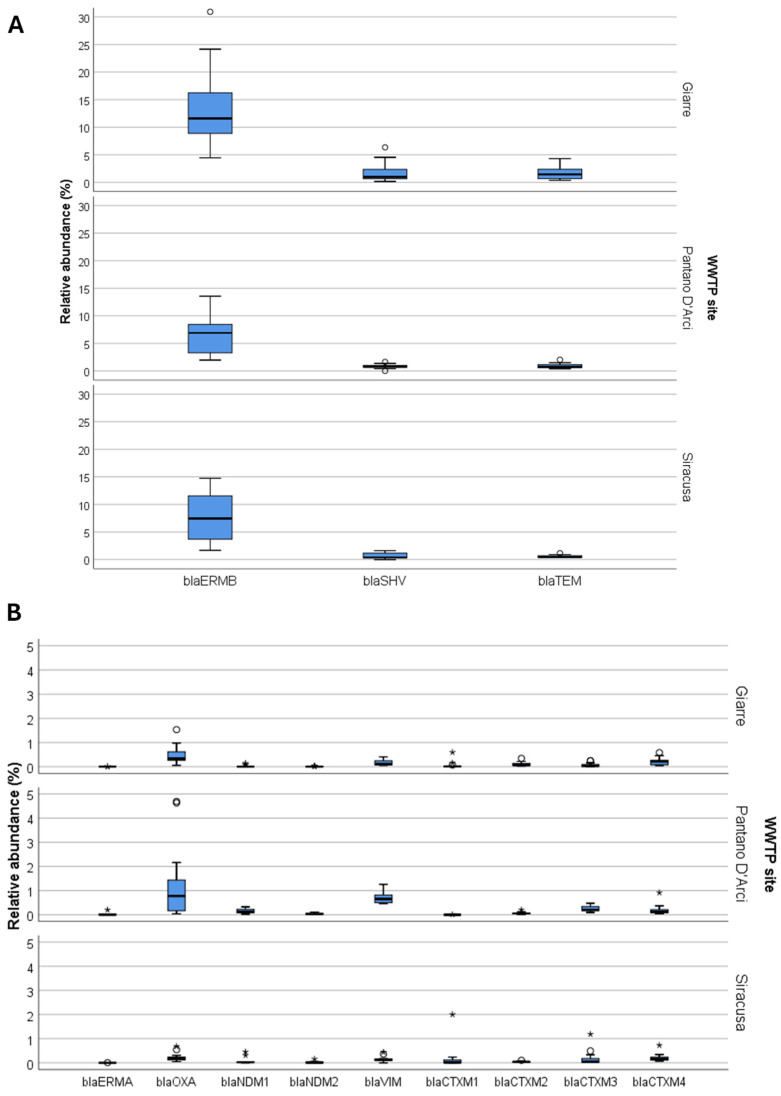
Relative abundances of ARGs in WWTP sites. (**A**) highlights the most abundant ARGs, including *erm(B)*, *bla*SHV, and *bla*TEM, while (**B**) displays ARGs with lower relative abundances. The figures present box plots of relative abundances for each ARG, with circles indicating potential outliers and asterisks representing extreme values.

**Figure 5 ijerph-22-00351-f005:**
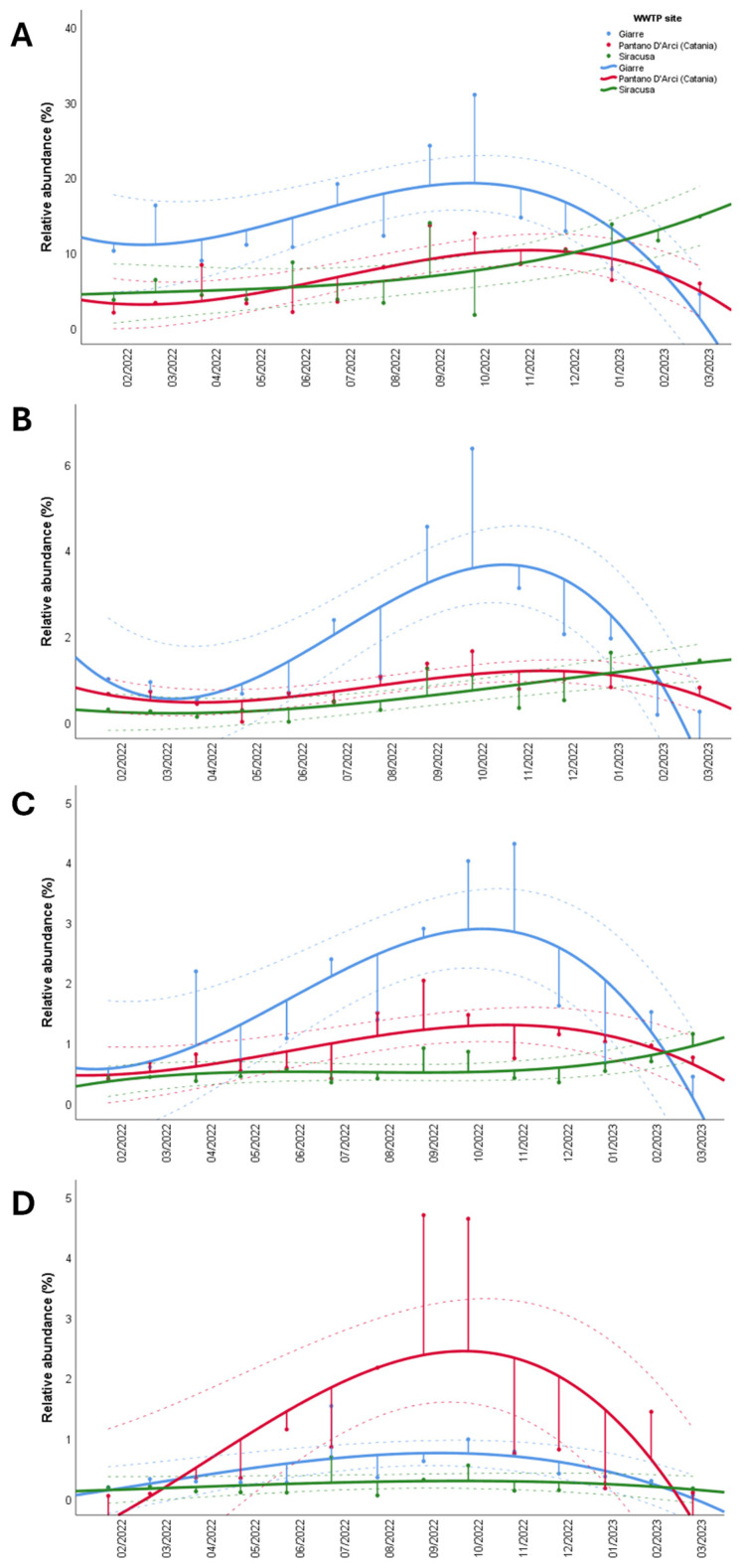
Temporal trends of ARGs’ relative abundances from February 2022 to March 2023. The panel shows results for *erm(B)* (**A**), *bla*SHV (**B**), *bla*TEM (**C**), and *bla*OXA (**D**). Each figure displays individual observations (points) and their projected distance from the trend line (vertical lines). The trend curves (solid line) and their corresponding confidence intervals (dashed lines) were derived using cubic fitting.

## Data Availability

Data are available from the corresponding authors upon reasonable request.

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
