# Peer review of "Monitoring Antibiotic Resistance in Wastewater: Findings from Three Treatment Plants in Sicily, Italy"

_ijerph, 2025, doi:10.3390/ijerph22030351_

Round 1
Reviewer 1 Report
Comments and Suggestions for Authors
It is a well conducted study, Some improvements are required including
1. Elaborate the number of samples from each collection site/plant in abstract
2. Results should be more clearly presented in abstract using numbers or percentages
3. In abstract, how blaSHV is abundant if it is not present in all samples??
4. Its better to mention the reference in section number 2.2, 2.3 and 2,5.
5. In first paragraph of discussion, no reference in mentioned.
6. Why pre-treatment samples are only collected from sites?? You can collect post treatment samples also to observe the efficacy of treatment procedures.
7. Page No.8, Line 242 The bacterial species/ genera should be italicised
8. Page No.7, Lines 216, 220, 226, 227, 231 The bacterial species/ genera should be italicised.
9. Figure 5 should be in prominent colors
10. In discussion its better to compare the results with some previous findings for better understanding of readers.
Author Response
Reviewer 1
It is a well conducted study, some improvements are required including
Comment (C): Elaborate the number of samples from each collection site/plant in abstract
Answer (A): Thank you for your thoughtful comment. As suggested, we added the number of samples from each collection site/plant in the revised version of the abstract.
C: Results should be more clearly presented in abstract using numbers or percentages
A: Thank you for your thoughtful comment. Please consider the revised version of our abstract.
C: In abstract, how blaSHV is abundant if it is not present in all samples??
A: We apologize if the abstract looked at is incomplete. The blaSHV gene is detected in 95.2% of samples and, when present, it showed higher relative abundance if compared with other genes of resistance.
C: Its better to mention the reference in section number 2.2, 2.3 and 2,5.
A: Thank you for your thoughtful comment. Please consider the revised version of our methods with the references in section number 2.2, 2.3 and 2,5.
C: In first paragraph of discussion, no reference in mentioned.
A: We apologize if the discussion looked at is incomplete. As requested, in the first paragraph, we have added the reference, which is the same one used to comment on the blaOXA gene.
C: Why pre-treatment samples are only collected from sites?? You can collect post treatment samples also to observe the efficacy of treatment procedures.
A: Thank you for your thoughtful comment. It was not possible to collect post-treatment samples in the early phase of the study; therefore, we have included among the limitations the inability to compare pre- and post-treatment samples to assess the effectiveness of the treatment plant. However, further studies are ongoing and will provide this information in the future.
C: Page No.8, Line 242 The bacterial species/ genera should be italicised
A: Thank you for your thoughtful comment. Please consider the revised version of our manuscript
C: Page No.7, Lines 216, 220, 226, 227, 231 The bacterial species/ genera should be italicised.
A: Thank you for your thoughtful comment. Please consider the revised version of our manuscript
C: Figure 5 should be in prominent colors
A: As suggested, we have revised Figure 5.
C: In discussion its better to compare the results with some previous findings for better understanding of readers.
A: We are grateful for this comment. To our knowledge, this is the first study aimed to assess the relative abundance of resistance genes in wastewater samples with the described detection method. Therefore, it was not possible to compare our results with previous studies published in the scientific literature. As suggested in the discussion, further studies are needed to compare the obtained results, also considering the specific characteristics of the treatment plants analysed.
Reviewer 2 Report
Comments and Suggestions for Authors
Lines 73 to 83: please provide more details of sampling, including sample triplicates, number of samples collected from each site, sampling dates and times, sampling type of grab versus composite, locations of the plants, maps of the sewersheds that the plants cover. Figure 1 is really unprofessional, not publishable, and unacceptable with a screen cut from Google maps, as per the high-standard journal quality of IJERPH. GIS maps of the sewersheds covered by the plants are needed for readers and scientific community to understand the areas covered by the plants. Re-creation of the Figure 1 is needed to demonstrate the sampling location (latitude and longitude), coverage area (sewersheds), jurisdiction boundaries, and more.
Line 86: why did the authors start with 45 mL of each sample at 56 degree? If it was a pre-developed protocol, please provide reference information.
Line 106: “samples collected weekly”, on which day did the samples collected during the week for each WWTP?
Lines 106 and 107: please provide references to support the approach of combining 4 weeks samples into a composite sample of a month. How many replicates or triplicates of each week’s sample? Section 2.4 needs more explanations of doing so. Why did not the authors analyze the weekly samples? Why did they want to combine 4 weeks samples into a composite sample? The time spanning of weekly samples over a month is long in terms of combining into a composite sample.
Lines 145 and 147: what are the demographic characteristics of the population covered? A table is needed to show the necessary parameters of demographic characteristics.
Figure 2: Since the high relative abundance of blaERM-B was observed across the 3 locations, other ARGs are not highlighted. Please separate the ARGs with high relative abundance such as blaERM-B and blaTEM with other ARGs with low relative abundance and create two or more separate graphs in terms of different levels of relative abundance of ARGs.
Figures 3 and 4: BlaERMB presented significantly higher relative abundance comparing to others and had an outlier. Add secondary y axis to fix them.
What is the y axis for Figure 5?
Lines 194 and 205 only described the trends of the ARGs but did not provide any insights of why these trends occurred in different times of the year. Please explain the seasonal variations of the relative abundance of the ARGs.
Overall, without addressing the comments above, I would be hesitating to move forward with this manuscript. Significant amount of information, especially sampling and study areas, is missing; the quality of figures 1 and 5 are poor.
Author Response
Reviewer 2
C: Lines 73 to 83: please provide more details of sampling, including sample triplicates, number of samples collected from each site, sampling dates and times, sampling type of grab versus composite, locations of the plants, maps of the sewersheds that the plants cover. Figure 1 is really unprofessional, not publishable, and unacceptable with a screen cut from Google maps, as per the high-standard journal quality of IJERPH. GIS maps of the sewersheds covered by the plants are needed for readers and scientific community to understand the areas covered by the plants. Re-creation of the Figure 1 is needed to demonstrate the sampling location (latitude and longitude), coverage area (sewersheds), jurisdiction boundaries, and more.
A: As suggested, the revised version of our manuscript includes more details on sampling. Moreover, we have also provided an improved Figure 1, representing sampling location, coverage area, and jurisdiction boundaries.
C: Line 86: why did the authors start with 45 mL of each sample at 56 degree? If it was a pre-developed protocol, please provide reference information.
A: Thank you for your thoughtful comment. Please consider the revised version of our manuscript in which we have provided reference information.
C: Line 106: “samples collected weekly”, on which day did the samples collected during the week for each WWTP?
A: Thank you for your comment. In the revised version of the method section, we have specified that the wastewater samples were collected at the beginning of each week for practical and logistical reasons.
C: Lines 106 and 107: please provide references to support the approach of combining 4 weeks samples into a composite sample of a month. How many replicates or triplicates of each week’s sample? Section 2.4 needs more explanations of doing so. Why did not the authors analyze the weekly samples? Why did they want to combine 4 weeks samples into a composite sample? The time spanning of weekly samples over a month is long in terms of combining into a composite sample.
A: Thank you for your valuable comment. We collected only one sample per week and then created a composite sample by combining the four weekly samples of the month. This composite sample was subsequently analysed in triplicate. We acknowledge the importance of clarifying this approach, and we have now provided additional details in Section 2.4. The decision to combine the weekly samples into a single monthly composite sample was made to obtain a more stable and integrated representation of the variations in the presence of antibiotic resistance genes (ARGs) over time, reducing the impact of potential daily or weekly fluctuations. This approach improves data representativeness and reduces the overall number of analyses while maintaining high reliability of the results. Anyway, we acknowledge limitations of using composite samples that are now described in the discussion section.
C: Lines 145 and 147: what are the demographic characteristics of the population covered? A table is needed to show the necessary parameters of demographic characteristics.
A: Please consider the revised version of our results, in which we added demographic characteristics of the population.
C: Figure 2: Since the high relative abundance of blaERM-B was observed across the 3 locations, other ARGs are not highlighted. Please separate the ARGs with high relative abundance such as blaERM-B and blaTEM with other ARGs with low relative abundance and create two or more separate graphs in terms of different levels of relative abundance of ARGs.
A: Thank you for your; we appreciate your feedback. However, we believe that dividing the heatmap into separate subplots is not the best approach. The purpose of this figure is to provide a comprehensive overview of the results, allowing for a direct comparison of ARGs across all locations. Additionally, the relative abundance values are clearly displayed within each cell, enabling readers to identify differences effectively. Separating the data into multiple graphs might fragment the overall interpretation and reduce the ability to observe broader trends across ARGs.
C: Figures 3 and 4: BlaERMB presented significantly higher relative abundance comparing to others and had an outlier. Add secondary y axis to fix them.
A: As suggested, we have provided two separate graphs for Figures 3 and 4.
C: What is the y axis for Figure 5?
A: We apologize for the mistake. Please consider the revised version of Figure 5.
C: Lines 194 and 205 only described the trends of the ARGs but did not provide any insights of why these trends occurred in different times of the year. Please explain the seasonal variations of the relative abundance of the ARGs. Overall, without addressing the comments above, I would be hesitating to move forward with this manuscript. Significant amount of information, especially sampling and study areas, is missing; the quality of figures 1 and 5 are poor.
A: Thank you for your insightful comment. We appreciate the opportunity to provide a possible explanation for the observed seasonal variations in the relative abundance of ARGs. The peaks in ARG abundance observed in September and October 2022 across all sites may be influenced by several environmental and anthropogenic factors. This peak may reflect increased antibiotic consumption during certain times of the year, such as the winter months when respiratory infections and other bacterial infections typically lead to higher antibiotic prescription. Alternatively, fluctuations in ARG levels may also correlate with population density changes, such as those seen in areas with seasonal tourism, which could temporarily increase the bacterial load in wastewater. During the study period, erm(B) showed higher relative abundances in Catania and Giarre, while Siracusa experienced an increase from September to March 2023. Similarly, blaSHV levels rose at all sites in September and October 2022, with Siracusa maintaining a steady trend until a slight increase in early 2023. For blaTEM, a peak was observed in Catania in September, followed by a gradual decline. In Siracusa, levels remained stable before increasing in March 2023, while in Giarre, the highest abundance was recorded in November 2022. As for blaOXA, its prevalence increased in Catania between August and October 2022, whereas Giarre showed a similar but smaller peak in July and October. In contrast, Siracusa exhibited a relatively stable pattern throughout the period. These findings suggest that climatic conditions, wastewater composition, and possibly fluctuations in antibiotic use could be driving the temporal trends observed. However, further studies incorporating environmental and clinical data would be necessary to confirm these hypotheses, as suggested in the discussion section.
Round 2
Reviewer 2 Report
Comments and Suggestions for Authors
The authors addressed my comments. Thank you.